# Differentiation of Benign and Malignant Breast Lesions Using ADC Values and ADC Ratio in Breast MRI

**DOI:** 10.3390/diagnostics12020332

**Published:** 2022-01-27

**Authors:** Silvia Tsvetkova, Katya Doykova, Anna Vasilska, Katya Sapunarova, Daniel Doykov, Vladimir Andonov, Petar Uchikov

**Affiliations:** 1Department of Diagnostic Imaging, Medical University Plovdiv, 15-A Vasil Aprilov Blvd, 4002 Plovdiv, Bulgaria; katya.doykova@mu-plovdiv.bg (K.D.); anna.vasilska@mu-plovdiv.bg (A.V.); 2First Department of Internal Diseases, Medical University Plovdiv, 15-A Vasil Aprilov Blvd, 4002 Plovdiv, Bulgaria; katya.sapunarova@mu-plovdiv.bg; 3Second Department of Internal Diseases, Medical University Plovdiv, 15-A Vasil Aprilov Blvd, 4002 Plovdiv, Bulgaria; daniel.doykov@mu-plovdiv.bg (D.D.); vladimir.andonov@mu-plovdiv.bg (V.A.); 4Department of Special Surgery, Medical University Plovdiv, 15-A Vasil Aprilov Blvd, 4002 Plovdiv, Bulgaria; puchikov@yahoo.com

**Keywords:** breast MRI, DWI, ADC value, ADC ratio, breast lesions

## Abstract

Magnetic resonance imaging (MRI) of the breast has been increasingly used for the detailed evaluation of breast lesions. Diffusion-weighted imaging (DWI) gives additional information for the lesions based on tissue cellularity. The aim of our study was to evaluate the possibilities of DWI, apparent diffusion coefficient (ADC) value and ADC ratio (the ratio between the ADC of the lesion and the ADC of normal glandular tissue) to differentiate benign from malignant breast lesions. Materials and methods: Eighty-seven patients with solid breast lesions (52 malignant and 35 benign) were examined on a 1.5 T MR scanner before histopathological evaluation. ADC values and ADC ratios were calculated. Results: The ADC values in the group with malignant tumors were significantly lower (mean 0.88 ± 0.15 × 10^−3^ mm^2^/s) in comparison with the group with benign lesions (mean 1.52 ± 0.23 × 10^−3^ mm^2^/s). A significantly lower ADC ratio was observed in the patients with malignant tumors (mean 0.66 ± 0.13) versus the patients with benign lesions (mean 1.12 ± 0.23). The cut-off point of the ADC value for differentiating malignant from benign breast tumors was 1.11 × 10^−3^ mm^2^/s with a sensitivity of 94.23%, specificity of 94.29%, and diagnostic accuracy of 98%, and an ADC ratio of ≤0.87 with a sensitivity of 94.23%, specificity of 91.43%, and a diagnostic accuracy of 95%. Conclusion: According to the results from our study DWI, ADC values and ADC ratio proved to be valuable additional techniques with high sensitivity and specificity for distinguishing benign from malignant breast lesions.

## 1. Introduction

Breast soft tissue lesions often represent a diagnostic challenge in daily practice. Covering a wide spectrum of histological conditions, they are broadly subdivided into malignant and benign tumor lesions. Breast cancer is the most common malignancy in the female population [1,2] and was even reported to be the most common malignancy in the world for both sexes in 2020, accounting for 11,7% of all newly diagnosed malignancies, closely followed by lung cancer (11.4%) [2]. Early diagnosis, including interventional procedures for histological confirmation, is crucial for a better outcome of the disease. On the other hand, following histological examination, lesions in the breast are frequently confirmed to be benign in origin, so unnecessary biopsies are often reported [3,4]. In this aspect, applying a non-invasive imaging technique with a high diagnostic potential is of great importance to avoid unnecessary interventional procedures and reduce costs, solving the dilemma of whether it is a malignant or non-malignant lesion [5,6,7]. Magnetic resonance imaging (MRI) of the breast has been used increasingly during the last decade as a preferred problem-solving method in complicated and unclear cases such as in women with “dense breast”, multiple lesions, the evaluation of residual tumor, recurrence or granulation tissue after intervention [8,9,10], due to its ability to reveal both the morphologic structure and the kinetic properties of the pathologic lesion [8,11].

Diffusion-weighted imaging (DWI) is a fast non-contrast MRI technique based on the free movement of water molecules in the extracellular space and reflects tissue cellularity. The motion of water molecules is more restricted in tissues with high cellularity and less restricted in areas of low cellularity [12,13]. Based on this characteristic, DWI can create contrast images that differ from the conventional T1- and T2-weighted images. Signal intensity in diffusion-weighted imaging is inversely proportional to the degree of water molecule diffusion [14], which means that structures with high cellularity and restriction in diffusion will present with a more intense signal. Furthermore, DWI allows the quantitative evaluation of water diffusion, using the apparent diffusion coefficient (ADC). Its value is calculated in square millimeters per second (mm^2^/s) and can be measured by assessing the signal attenuation that occurs at diffusion-weighted imaging performed with at least two different b values [14]. Breast cancer usually presents with a restricted diffusion of water molecules which leads to increased DWI signal. The ADC value is lower as compared to normal breast tissue and benign lesions of the breast [14,15]. High ADC values are rarely reported in malignant lesions [16,17]. Some authors additionally used the ADC ratio between the ADC value of the lesion and ADC value of normal breast glandular tissue to further evaluate the diagnostic performance of DWI [18,19].

However, data in the literature on the diagnostic potential of the applied technique show some discrepancies, not only in terms of its sensitivity and specificity. Studies have also demonstrated some inconclusiveness regarding the ADC value threshold that could be trusted when attempting to non-invasively differentiate malignant from benign breast lesions [20,21].

Based on the existing discrepancies, in our study, we aimed to evaluate the role of DWI, ADC value and ADC ratio in the differentiation of benign from malignant breast lesions in patients with proven histopathological diagnoses.

## 2. Materials and Methods

### 2.1. Patients

This retrospective observational study includes 87 patients with solid breast lesions diagnosed by mammography or breast ultrasound that needed further assessment. They were referred to breast MRI during the period of January 2018–July 2021. All MRI examinations were performed prior to biopsy procedures. The absence of subsequent biopsy and histological findings was the main exclusion criterion. Nine patients had a Tru-cut biopsy, while 78 patients underwent excisional biopsy. The type of biopsy was decided by the surgeon. Tru-cut biopsy was chosen as a less invasive method as compared to excisional biopsy and was performed using a Tru-cut gun with an 18-gauge needle. The tissue specimen included four consecutive insertions of the needle in the lesion, while excisional biopsy usually removed the whole lesion or pathologic region. All patients signed written informed consent for all procedures.

### 2.2. MRI Protocol

All MRI examinations were acquired on a 1.5 T MRI scanner (Magnetom Amira, Siemens Healthcare, Erlangen, Germany) using an 18-channel dedicated breast coil. Patients were examined in a prone position, head first. The standardized MRI protocol included the following sequences: non-fat-suppressed T1-weighted transversal sequence; fat-suppressed turbo inversion recovery magnitude (TIRM) transversal sequence; fat-suppressed dynamic contrast-enhanced 3D T1—weighted fast low angle (FL) transversal sequence. A gadolinium contrast agent was injected (0.1 mmol/kg) and 1 pre-contrast and 6 post-contrast series were performed with a slice thickness of 1.5 mm. Subtraction images were also acquired by subtracting the post-contrast images from the first images. Time signal intensity curves of the lesions were also obtained from the dynamic series. As a basic sequence of evaluation in the study, DWI was performed prior to the dynamic contrast examination with the following parameters: field of view (FOV) 420 × 200 mm, time of repetition (TR) 5400 ms, TE 53 ms, matrix size 160 × 61, slice thickness 4 mm, and scanning time 232 s. The diffusion-weighted sequences were performed in the axial plane with 3 B values (B = 50, 500, 800 mm^2^/s). ADC maps were received during the examination and ADC values were calculated using software provided by the manufacturer (Syngo, Siemens Healthcare, Erlangen, Germany).

### 2.3. ADC Value Measurement

ADC values of the lesions were measured by manually placing regions of interest (ROIs) within the lesion on the ADC map. Multiple ROIs were placed and the ROI with the lowest ADC value was selected. We avoided central regions with necrosis and obvious cystic areas within the lesion as well as the most peripheral areas to avoid partial volume effects. Areas with obvious artifacts were also avoided. When identifying the solid part of the tumors, we used as reference images those from the dynamic contrast-enhanced MRI (DCE MRI), usually the third post-contrast series and the subtraction image. The ADC value of normal glandular tissue was assessed on the contralateral breast except for one patient who had previous mastectomy. After that, we calculated the ADC ratio which is the ratio between the ADC of the lesion and ADC of the normal glandular tissue. The assessment was performed by one qualified radiologist with 4 years of experience in breast MRI.

### 2.4. Pathohistological Evaluation

All 87 patients had a final diagnosis based on histological examination obtained after Tru-cut or excisional biopsy. The pathohistological report assessed the lesion type (benign or malignant) and detailed histological findings.

### 2.5. Statistical Analysis

The data analysis was performed with the statistical software IBM SPSS version 27 (Chicago, IL, USA, 2020) and MedCalc version 20.014 (MedCalc Software Ltd., Ostend, Belgium, 2021). Continuously measured variables (age, ADC, ADC ratio) were normally distributed (Kolmogorov–Smirnov’s *p* > 0.05 for all variables) and were described through the mean values and standard deviations. These variables were compared between the patients with malignant and benign tumors through t-tests for independent samples. A receiver operating characteristic (ROC) curve was used to establish the cut-off ADC value and ADC ratio for distinguishing malignant from benign tumors with the corresponding levels of sensitivity and specificity. All statistical tests were two-tailed and performed at a level of significance alpha = 0.05. The statistical significance was marked as follows: *—*p* < 0.05; **—*p* < 0.01; ***—*p* < 0.001. 

## 3. Results

This study included 87 patients with a mean age of 48.05 ± 8.23 years (range 21 to 60 years). All patients had a breast MRI including both DCE MRI and DWI. Based on the histological evaluation (Table 1), 52 (59.76%) of the patients were diagnosed with malignant tumors, of which the most frequent type (*n* = 36) was invasive ductal carcinoma (IDC). The remaining 35 (40.24%) patients had benign lesions, among which the most frequent type was fibroadenoma (*n* = 12). The mean age of the patients with malignant tumors was significantly higher (49.79 ± 7.87 years, range of 24–60 years old) in comparison to the patients with benign lesions (45.45 ± 8.17 years, range of 21–60 years old), *p* = 0.015.

The ADC values in the group with malignant tumors were significantly lower (mean 0.88 ± 0.15 × 10^−3^ mm^2^/s; range 0.60 to 1.30 × 10^−3^ mm^2^/s) in comparison with the group with benign lesions (mean 1.52 ± 0.23 × 10^−3^ mm^2^/s; range 0.99 to 2.01 × 10^−3^ mm^2^/s), *p* < 0.001. One of the 52 malignant tumors showed an ADC value of 1.30 × 10^−3^ mm^2^/and it was ductal carcinoma in situ (DCIS), and one of the 35 benign lesions (an abscess) had an ADC value of 0.89 × 10^−3^ mm^2^/s (Figure 1).

The analysis with the receiver operating characteristic (ROC) curve determined ADC ≤ 1.11 ×10^−3^ mm^2^/s as the optimum cut-off value distinguishing malignant from benign tumors with a sensitivity of 94.23%, a specificity of 94.29%, and a very high diagnostic accuracy of 98% (AUC = 0.981, 95%CI: 0.928 to 0.998, *p* < 0.001) (Figure 2).

For normal fibroglandular tissue, the ADC values in the studied group of patients ranged between 0.99 and 1.72 × 10^−3^ mm^2^/s, with a mean of 1.36 ± 0.16 × 10^−3^ mm^2^/s. A significantly lower ADC ratio was observed in the patients with malignant tumors (mean of 0.66 ± 0.13, range of 0.43–1.06) versus the patients with benign lesions (mean 1.12 ± 0.23, range 0.54 to 1.79), *p* < 0.001 (Figure 3).

An ADC ratio cut-off point of ≤0.87 was established as the optimum criterion for differentiating malignant from benign breast tumors, characterized by 94.23% sensitivity, 91.43% specificity, and a diagnostic accuracy of 95% (AUC = 0.950, 95% CI: 0.881 to 0.985, *p* < 0.001) (Figure 4).

## 4. Discussion

MRI is a preferred problem-solving imaging method for the evaluation of complex and unclear breast lesions in women with “dense breast” or multiple lesions, in the evaluation of residual tumor, recurrence or granulation tissue due to its ability to assess both the morphologic structure and the kinetic properties of the pathologic lesions [8,9,10,11]. However, MRI has a high sensitivity but lower specificity (93% and 71%, respectively) [21] in the assessment of breast lesions. Recent studies [20,21,22,23,24] have suggested that the additional evaluation of the diffusion properties of breast lesions can improve specificity; ADC, measured in DWI, is thus being increasingly used as a marker in the detection and characterization of breast lesions [21]. Furthermore, other authors added to this additional ADC ratio calculation [25,26]. DWI and ADC values are determined by the decrease in the extracellular volume content of the tumors due to the increased cellular density and the fall in the ADC value is due to restricted water diffusion [26]. This is different structural information than that from the dynamic contrast-enhanced examinations whose results are directly related to the vascularity of the tumors and unrelated to tumor cellularity [26,27].

Our study evaluates the ADC values and ADC ratio of breast lesions in patients with benign and malignant tumors before the performance of biopsy procedures. The ADC values showed a significant difference (*p* < 0.001) between malignant and benign lesions. In the group with malignant tumors, they were significantly lower (mean 0.88 ± 0.15 × 10^−3^ mm^2^/s; range 0.60 to 1.30 × 10^−3^ mm^2^/s) in comparison with the group with benign lesions (mean 1.52 ± 0.23 × 10^−3^ mm^2^/s; range 0.99 to 2.01 × 10^−3^ mm^2^/s). Our results for malignant tumors are compatible with the reported range of mean ADC values in the literature varying from 0.83 ± 0.19 × 10^−3^ mm^2^/s to 1.52 ± 0.23 × 10^−3^ mm^2^/s [18,28,29]. However, our results were closer to the results of Kim et al. [22] that reported ADC values of 0.87–0.93 × 10^−3^ mm^2^/s and Akin et al. [25]—0.83 ± 0.19 × 10^−3^ mm^2^/s, which are higher than those reported by Maric et al. [23]—0.68 × 10^−3^ mm^2^/s and lower than those of Partridge et al. [28]—1.29 ± 0.26 × 10^−3^ mm^2^/s. These results may be due to the different technical parameters and different histological distribution in the studies. In our study, the lowest value 0.60 × 10^−3^ mm^2^/s was in a 45-year-old patient with invasive ductal carcinoma (Figure 5).

The ADC values of benign tumors are also within the reported values from 1.41 ± 0.24 × 10^−3^ mm^2^/s to 1.72 ± 0.43 × 10^−3^ mm^2^/s [23,25,30,31].

Different studies reported varying results in ADC value sensitivity and specificity. We determined an ADC ≤ 1.11 × 10^−3^ mm^2^/s as the optimum cut-off value for distinguishing malignant from benign tumors with a sensitivity of 94.23%, a specificity of 94.29%, and a very high diagnostic accuracy of 98% (AUC = 0.981, 95%CI: 0.928 to 0.998, *p* < 0.001). A similar cut-off value of 1.1 × 10^−3^ mm^2^/s was reported by Azab and Ibrahim [18], achieving a sensitivity of 89.75% and a specificity of 94.4%, Akin et al. [25] 1.08 × 10^−3^ mm^2^/s with sensitivity and specificity of 92% and 92%, respectively, while Kul et al. [32] reported a cut-off value of 0.92 × 10^−3^ mm^2^/s with a sensitivity of 91.5% and a specificity of 86.5%. A meta-analysis based on 13,847 lesions from 123 studies, published in 2019, established that an ADC cut-off value of 1.00 × 10^− 3^ mm^2^/s can be recommended for distinguishing breast cancers from benign lesions. This result was independent on Tesla strength, choice of b values, and measure methods (whole lesion measure vs. estimation of ADC in a single area) [33]. In another prospective multicenter study of 107 women with MRI-detected BI-RADS 3, 4, or 5 lesions, Rahbar et al. [34] evaluated the diagnostic performance of centrally measured ADC values to identify optimal ADC thresholds to reduce unnecessary biopsies. They identified an ADC threshold of 1.53 × 10^−3^ mm^2^/s, which lowered the biopsy rate by 20.9%. The authors recommended that an established threshold should be validated in future studies. In 2021, Clauser et al. [7] conducted a retrospective, multicentric, cross-sectional study in five sites in three European countries to evaluate whether the pre-defined ADC cut-off value by Rahbar et al. [34] allows the downgrading of BI-RADS 4 lesions on contrast-enhanced MRI, thus avoiding unnecessary biopsies. This study included 657 female patients with 696 BI-RADS 4 lesions. Applying the investigated ADC cut-off, sensitivity was 96.6% and the potential reduction in unnecessary biopsies was found to be 32.6%.

In our study, with the application of ADC ≤ 1.11 × 10^−3^ mm^2^/s as the optimum cut-off value distinguishing malignant from benign tumors, we had two malignant lesions that showed “false-negative results”, which are both DCIS with ADC values, respectively, 1.3 and 1.21 × 10^−3^ mm^2^/s, and two lesions which were just at the cut-off values of inflammatory carcinoma and mucinous carcinoma, respectively.

We also had one “false-positive result”—an abscess with an ADC value of 0.89 × 10^−3^ mm^2^/s (Figure 6) and another abscess at the cut-off value of 1.11 × 10^−3^ mm^2^/s. The highest ADC value was for fibroadenoma—2.01 × 10^−3^ mm^2^/s.

Additionally, we calculated the ADC ratio by dividing the ADC value of the lesion and the ADC value of the normal glandular tissue. A significantly lower ADC ratio was observed in the patients with malignant tumors (mean 0.66 ± 0.13, range 0.43 to 1.06) versus the patients with benign lesions (mean 1.12 ± 0.23, rang 0.54 to 1.79), *p* < 0.001. A cut-off point of ADC ratio ≤ 0.87 was established as the optimum criterion for differentiating malignant from benign breast tumors, characterized by 94.23% sensitivity, 91.43% specificity, and a diagnostic accuracy of 95% (AUC = 0.950, 95% CI: 0.881 to 0.985, *p* < 0.001). In our study, the calculated ADC ratio did not additionally improve the results received when using ADC values only. Several studies in the literature have reported increased sensitivity and specificity when using the ADC ratio: Azab and Ibrahim [18] reported a cut-off ADC ratio of 0.9 in the differentiation between benign and malignant breast lesions with a sensitivity of 92.2% and a specificity of 94.4% and Sahin and Aribal [35] reported a cut-off ADC ratio of 0.8 with a sensitivity of 91.4% and a specificity of 100%.

Although we believe that the data from our study are valuable, we must mention some limitations. First, this study was retrospective and existing artifacts could not be corrected. Second, the data were obtained and analyzed by one radiologist and in one center. Third, the distribution of patients in the different histological subgroups showed that invasive ductal carcinoma was dominant in the malignant group and fibroadenoma in the benign group. Thus, due to the small number of patients, we were unable to perform a subgroup analysis and test the relation between the ADC value and ADC ratio in different benign and malignant subgroups. This could give us more information in relation to the biological nature of the lesions, which are worth being in the scope of future investigations.

Despite the aforementioned limitations, we believe that our study presents valuable data about ADC values, ADC ratios, and their potential for differentiating benign from malignant breast lesions—and thus adds to the existing knowledge.

## 5. Conclusions

In conclusion, the data from our study demonstrated the diagnostic potential of DWI, ADC value, and ADC ratio in differentiating malignant from benign breast lesions. Based on our data, we would propose the usage of an ADC value of ≤1.11 × 10^−3^ mm^2^/s as the optimum cut-off level to distinguish benign from malignant breast lesions. As the ADC ratio did not further improve the predictive potential and diagnostic accuracy of ADC values, we consider the ADC value as reliable as the ADC ratio. Both measurements could be implemented additionally to well-established imaging techniques to more accurately predict the nature of the breast lesions. This would help avoid unnecessary or even risky interventional diagnostic procedures. We realize that the ADC value around the threshold still presents a challenge and will require special attention when making a particular diagnostic decision. However, this does not lower the importance of the technique as a part of the comprehensive diagnostic evaluation of breast lesions.

## Figures and Tables

**Figure 1 diagnostics-12-00332-f001:**
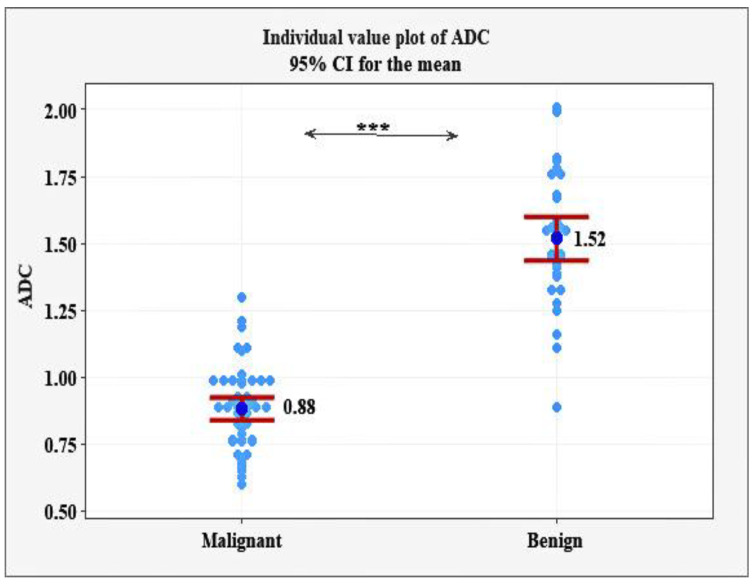
Individual and mean ADC values in the patients with malignant and benign tumors, *** *p* < 0.001.

**Figure 2 diagnostics-12-00332-f002:**
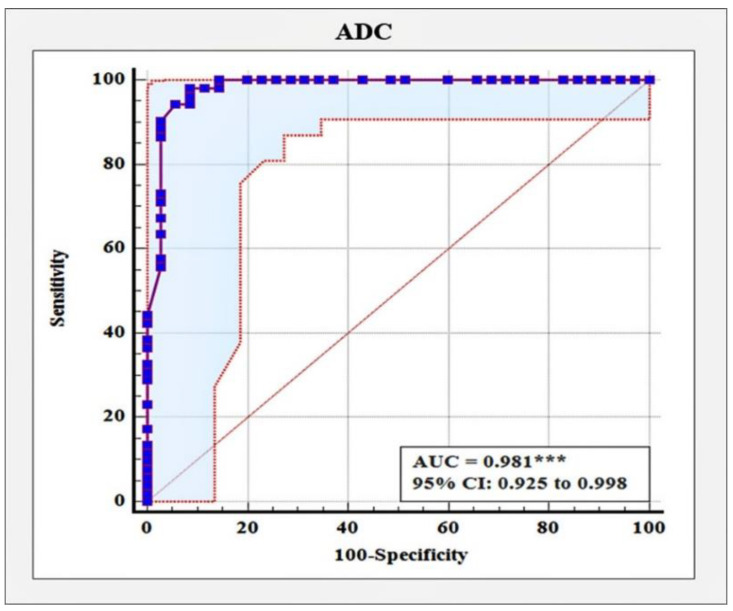
Receiver operating characteristic curve for distinguishing malignant from benign tumors based on ADC values, *** *p* < 0.001.

**Figure 3 diagnostics-12-00332-f003:**
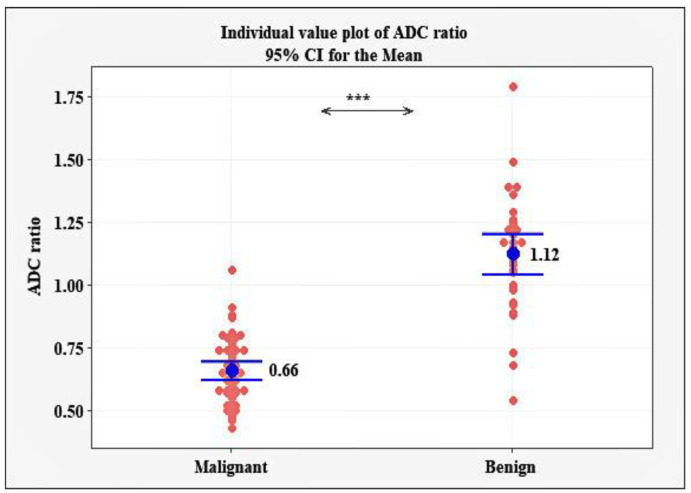
Individual and mean ADC ratio for the patients with malignant and benign tumors, *** *p* < 0.001.

**Figure 4 diagnostics-12-00332-f004:**
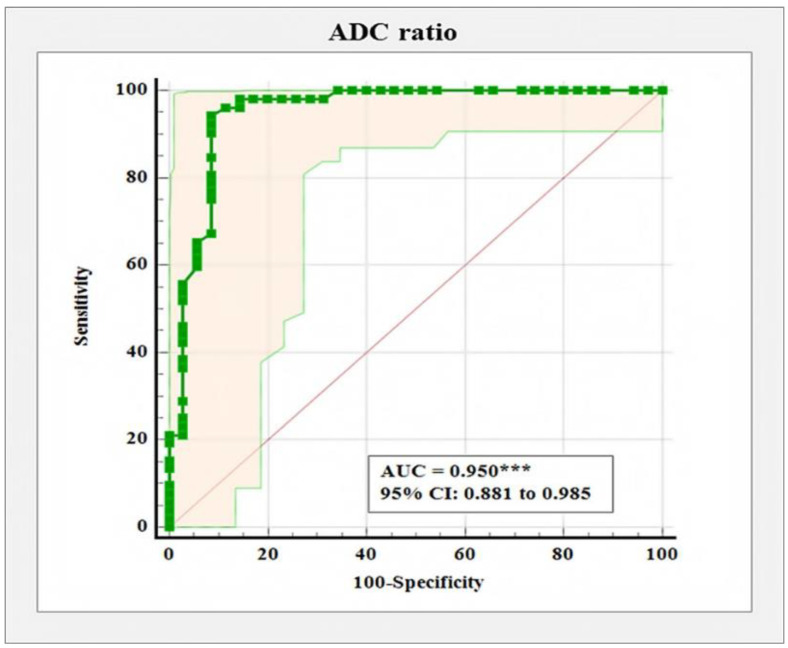
Receiver operating characteristic curve for distinguishing malignant from benign tumors based on ADC ratio, ****p* < 0.001.

**Figure 5 diagnostics-12-00332-f005:**
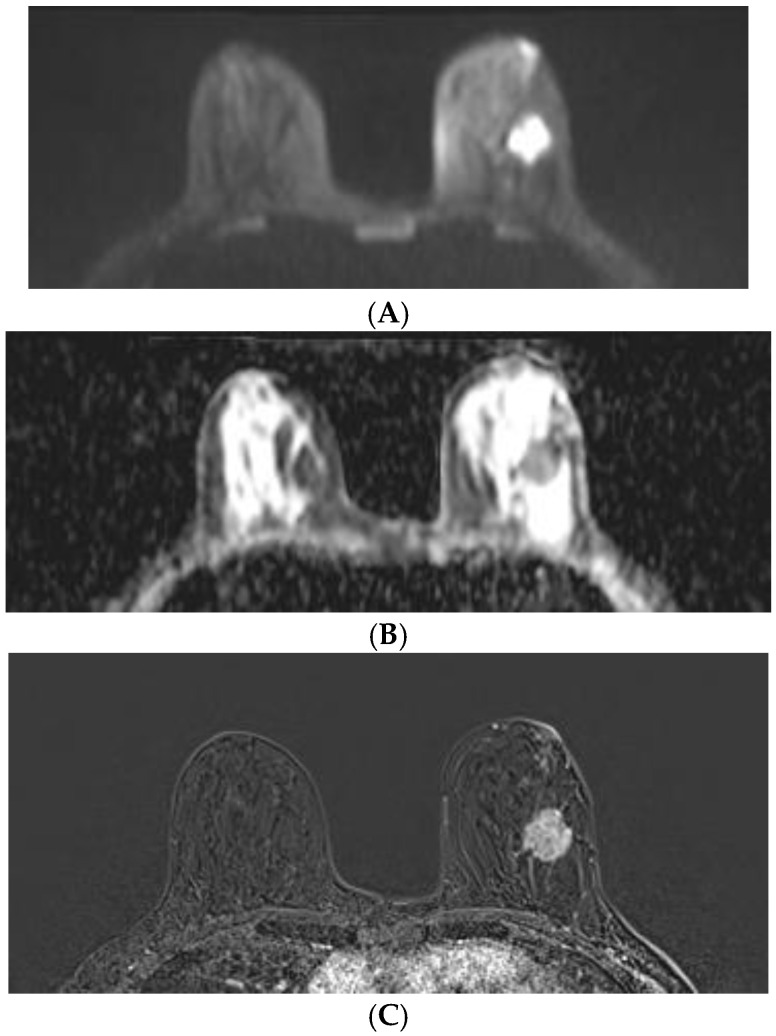
Breast MRI in a 45-year-old patient with invasive ductal carcinoma: (**A**) DWI (b = 800 mm^2^/s); (**B**) ADC value 0.60 × 10^−3^ mm^2^/s; and (**C**) DCE-MRI (subtraction image).

**Figure 6 diagnostics-12-00332-f006:**
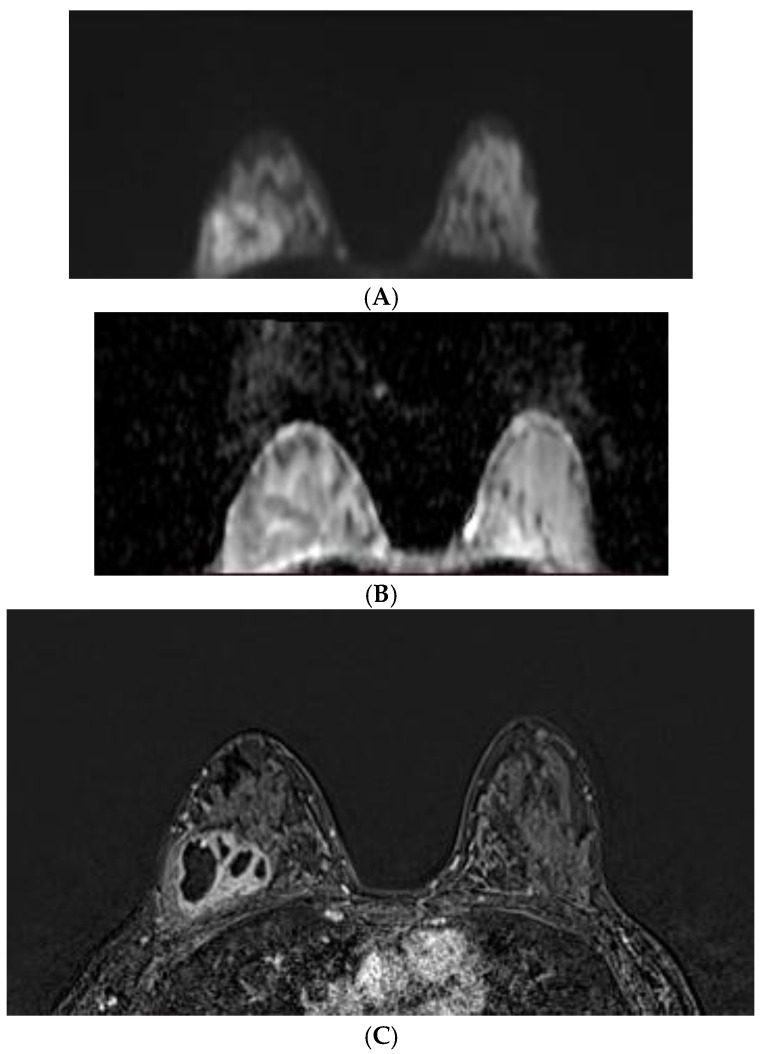
Breast MRI in a 29-year-old patient with an abscess: (**A**) DWI (b = 800 mm^2^/s); (**B**) ADC value 0.89 × 10^−3^ mm^2^/s; and (**C**) DCE-MRI (subtraction image).

**Table 1 diagnostics-12-00332-t001:** Histopathological data.

Histopathological Findings	Frequency	Percentage
Malignant		
○ Invasive ductal carcinoma	36	41.40
○ Invasive lobular carcinoma	5	5.75
○ Ductal carcinoma in situ (DCIS)	3	3.45
○ Inflammatory carcinoma	2	2.30
○ Medullary carcinoma	2	2.30
○ Anaplastic carcinoma	1	1.14
○ Mucinous carcinoma	1	1.14
○ Papillary carcinoma	1	1.14
○ Tubular carcinoma	1	1.14
**Total**	52	59.76%
Benign		
○ Fibroadenoma	12	13.8
○ Abscess	3	3.45
○ Apocrine metaplasia	3	3.45
○ Sclerosing adenosis	3	3.45
○ Fibrosis	3	3.45
○ Fibrocystic changes	3	3.45
○ Postoperative granular tissue	3	3.45
○ Intraductal papilloma	2	2.30
○ Fat necrosis	2	2.30
○ Inflamed cyst	1	1.14
**Total**	35	40.24%

## Data Availability

Not applicable.

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
