# Peer review of "Differentiation of Benign and Malignant Breast Lesions Using ADC Values and ADC Ratio in Breast MRI"

_diagnostics, 2022, doi:10.3390/diagnostics12020332_

Round 1
Reviewer 1 Report
The paper consists in the evaluation of the possibilities of DWI, apparent diffusion coefficient (ADC) value and ADC ratio to differentiate benign from malignant breast lesions. The paper is interesting and current, but the research is very superficial to be accepted.
The introduction must include more background and references. Also, the aim and scope of the study must be better presented in the introduction. Also, the results obtained must be introduced.
The methods are clearly described. However, the results must be detailed, and compared with the literature in a discussion section. It is not possible to infer the reliability of the study without comparison. The limitations must be better written.
The conclusions can have related work.
The paper needs a lot of improvements before its acceptance.
Author Response
Thank you very much for the review of our manuscript and the valuable advice. This article is very important for us and we are making our best to meet all the requirements of rewiewers. We have complied with all the comments made by reviewer 1 and have done the appropriate corrections. We believe that this will improve the quality of the presented article. Other additional changes have been made in the article, in accordance with the requirements of reviewer 2.
If you have additional comments and recommendations, please do not hesitate to contact us.
Point 1: The introduction must include more background and references. Also, the aim and scope of the study must be better presented in the introduction. Also, the results obtained must be introduced.
The introduction is revised, expanded with additional information, more background and references are included. The aim and scope of the study are more clearly stated, some of the results are introduced.
Point 2: The methods are clearly described. However, the results must be detailed and compared with the literature in a discussion section. It is not possible to infer the reliability of the study without comparison. The limitations must be better written.
All the results are presented in detail and the discussion is expanded with a comparison with the literature in more detail as well. The limitations of the study are revised and more clearly stated.
Point 3: The conclusions can have related work.
The conclusion part is revised and improved.
Reviewer 2 Report
In this study, authors have investigated the potential benefits of diffusion-weighted imaging to identify benign and malignant breast lesions. The manuscript is well prepared and structured. However, several issues need to be addressed for the reader’s benefit.
- The second statement of the abstract is too wide and may confuse. Please consider revising it.
- It’s better to describe the ADC ratio at first use in the abstract to prevent confusion.
- In the conclusion part of the abstract, the statement is too strong based on the size of the sample size. You may soften the statement.
- The first statement needs to be cited.
- Please consider revising the first sentence of the second paragraph.
- There was misconnection as starting to statement at the second line of the second page. Also, you can combine the hypothesis paragraph with the previous one.
- Please consider briefly introducing the tru-cut and excisional biopsy procedures.
- You may remove the second and third statements of the MRI protocol section.
- Could you please briefly mention the reason for computing the subtraction images?
- Why did you compute the ADC maps twice?
- Could you please describe the second sentence of the ADC values measurement part?
- What is the reason for using two statistics software?
- Could you please align Table 1 to left?
- Please check the ADC values, power sections.
- Please describe DCIS at first usage.
- Please replace the term “basic” in the first sentence of the discussion section
- Please check the statement starting with “Recent studies [6-8]”.
- The use of “DWI and ADC values” is misleading. Please check it.
- Please split the sentence starting with “The ADC values showed …”
- Please realign sub-sections of Figure 6.
- Please consider revising the term “estimated” in the first line of the third paragraph on page 7.
- Please cite the sentence starting with “Several studies in the literature …”
- Please check the format of the paragraph on page 8.
- Please consider revising the term “very accurate” since the patient cohort is considerably small.
- The last sentence is not directly related to the current study. Please revise it.
- One of the references is missing doi. Please include it.
Author Response
The authors are grateful for the review and the valuable advice. We have complied with all the comments made by the reviewer and have done the appropriate corrections. We believe that this will improve the quality of the presented article. Other additional changes have been made in the article, in accordance with the requirements of reviewer 1.
Point 1: The second statement of the abstract is too wide and may confuse. Please consider revising it.
Response 1: The second statement of the abstract is revised.
Point 2: It’s better to describe the ADC ratio at first use in the abstract to prevent confusion.
Response 2: The description of ADC ratio is included in the abstract.
Point 3: In the conclusion part of the abstract, the statement is too strong based on the size of the sample size. You may soften the statement.
Response 3: The statement is changed.
Point 4: The first statement needs to be cited.
Response 4: The statement is cited and additional information is included.
Point 5: Please consider revising the first sentence of the second paragraph.
Response 5: Revised
Point 6: There was misconnection as starting to statement at the second line of the second page. Also, you can combine the hypothesis paragraph with the previous one.
Response 6: Revised and combined.
Point 7: Please consider briefly introducing the tru-cut and excisional biopsy procedures.
Response 7: Information is included
Point 8: You may remove the second and third statements of the MRI protocol section.
Response 8: The third statement is removed.
Point 9: Could you please briefly mention the reason for computing the subtraction images?
Response 9: Subtraction images are not computed.
Point 10: Why did you compute the ADC maps twice?
Response 10: ADC maps were obtained during the examination on the MRI scanner and ADC values were estimated on the Singo workstation. Changed in the paper in the correct way.
Point 11: Could you please describe the second sentence of the ADC values measurement part?
Response 11: Multiple ROIs were placed and the RIO with the lowest ADC value was selected. Changed in the paper in the correct way.
Point 12: What is the reason for using two statistics software?
Response 12: The SPSS and MedCalc statistical programs were used for different tasks. The screening of the data, frequency analysis, descriptive statistics, and t-tests were performed through the SPSS program. MedCalc was used to perform the ROC curve analysis because being specifically developed for the medical field, MedCalc provides very detailed and specific results, which we find more informative as compared to those produced by SPSS.
Point 13: Could you please align Table 1 to left?
Response 13: Of course, we did it, thank you.
Point 14: Please check the ADC values, power sections.
Response 14: If I have understood correctly they were written without superscript, now it is corrected.
Point 15: Please describe DCIS at first usage
Response 15: Described.
Point 16: Please replace the term “basic” in the first sentence of the discussion section
Response 16: Replaced
Point 17: Please check the statement starting with “Recent studies [6-8]”.
Response 17: Revised
Point 18: The use of “DWI and ADC values” is misleading. Please check it.
Response 18: Checked and changed
Point 19: Please split the sentence starting with “The ADC values showed …”
Response 19: Corrected
Point 20: Please realign sub-sections of Figure 6.
Response 20: Realigned
Point 21: Please consider revising the term “estimated” in the first line of the third paragraph on page 7.
Response 21:Revised and replaced.
Point 22: Please cite the sentence starting with “Several studies in the literature …”
Response 22: Cited.
Point 23: Please check the format of the paragraph on page 8.
Response 23: Format corrected.
Point 24: Please consider revising the term “very accurate” since the patient cohort is considerably small.
Response 24: Revised
Point 25: The last sentence is not directly related to the current study. Please revise it.
Response 25: Revised
Point 26: One of the references is missing doi. Please include it.
Response 26: Doi included
Round 2
Reviewer 1 Report
The authors made sufficient corrections for the acceptance of the paper.